# CONNECTING THE DOTS BETWEEN MLE AND RL FOR SEQUENCE PREDICTION

## ABSTRACT

Sequence prediction models can be learned from example sequences with a variety of training algorithms. Maximum likelihood learning is simple and efficient, yet can suffer from compounding error at test time. Reinforcement learning such as policy gradient addresses the issue but can have prohibitively poor exploration efficiency. A rich set of other algorithms, such as data noising, RAML, and softmax policy gradient, have also been developed from different perspectives. In this paper, we present a formalism of entropy regularized policy optimization, and show that the apparently distinct algorithms, including MLE, can be reformulated as special instances of the formulation. The difference between them is characterized by the reward function and two weight hyperparameters. The unifying interpretation enables us to systematically compare the algorithms side-by-side, and gain new insights into the trade-offs of the algorithm design. The new perspective also leads to an improved approach that dynamically interpolates among the family of algorithms, and learns the model in a scheduled way. Experiments on machine translation, text summarization, and game imitation learning demonstrate superiority of the proposed approach.

## 1 INTRODUCTION

Sequence prediction problem is ubiquitous in many applications, such as generating a sequence of words for machine translation (Wu et al., 2016; Sutskever et al., 2014), text summarization (Hovy & Lin, 1998; Rush et al., 2015), and image captioning (Vinyals et al., 2015; Karpathy & Fei-Fei, 2015), or taking a sequence of actions to complete a task. In these problems (e.g., Mnih et al., 2015; Ho & Ermon, 2016), we are often given a set of sequence examples, from which we want to learn a model that sequentially makes the next prediction (e.g., generating the next token) given the current state (e.g., the previous tokens).

A standard training algorithm is based on supervised learning which seeks to maximize the log-likelihood of example sequences (i.e., maximum likelihood estimation, MLE). Despite the computational simplicity and efficiency, MLE training can suffer from compounding error (Ranzato et al., 2016; Ross & Bagnell, 2010) in that mistakes at test time accumulate along the way and lead to states far from the training data. Another line of approaches overcome the training/test discrepancy issue by resorting to the reinforcement learning (RL) techniques (Ranzato et al., 2016; Bahdanau et al., 2017; Rennie et al., 2017). For example, Ranzato et al. (2016) used policy gradient (Sutton et al., 2000) to train a text generation model with the task metric (e.g., BLEU) as reward. However, RL-based approaches can face challenges of prohibitively poor sample efficiency and high variance. To this end, a diverse set of methods has been developed that is in a middle ground between the two paradigms of MLE and RL. For example, RAML (Norouzi et al., 2016) adds reward-aware perturbation to the MLE data examples; SPG (Ding & Soricut, 2017) leverages reward distribution for effective sampling of policy gradient. Other approaches such as data noising (Xie et al., 2017) also show improved results.

In this paper, we establish a unifying perspective of the above distinct learning algorithms. Specifically, we present a generalized entropy regularized policy optimization framework, and show that the diverse algorithms, such as MLE, RAML, data noising, and SPG, can all be re-formulated as special cases of the framework, with the only difference being the choice of reward and the values of two weight hyperparameters (Figure 1). In particular, we show MLE is equivalent to using a

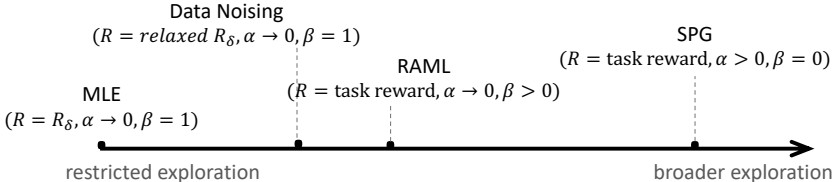

Figure 1: A unifying formulation of different learning algorithms. Each algorithm is a special instance of the generalized ERPO formalism (Eq.1), by using different rewards and taking different values of the weight hyperparameters $\alpha, \beta$.

*Delta*-function reward which returns 1 to model samples that match training examples exactly, and $-\infty$ to any other samples. Such extremely restricted reward has literally disabled any exploration of the model beyond training data, yielding brittle prediction behaviors. Other algorithms essentially use various locally-relaxed rewards, joint with the model distribution, for broader (and more costly) exploration during training.

Besides the new views of the existing algorithms, the unifying perspective also leads to new algorithms for improved learning. We develop *interpolation algorithm*, which, as training proceeds, gradually expands the exploration space by annealing both the reward function and the weight hyperparameters. The annealing in effect dynamically interpolates among the existing algorithms from left to right in Figure 1. We conduct experiments on the tasks of text generation including machine translation and text summarization, and game imitation learning. The interpolation algorithm shows superior performance over various previous methods.

## 2  RELATED WORK

Given a set of data examples, sequence prediction models are usually trained to maximize the log-likelihood of the next label (token, action) conditioning on the current state observed in the data. Reinforcement learning (RL) addresses the discrepancy between training and test by also using models' own predictions at training time. Various RL approaches have been applied for sequence generation, such as policy gradient (Ranzato et al., 2016) and actor-critic (Bahdanau et al., 2017). Reward augmented maximum likelihood (RAML) (Norouzi et al., 2016) is an algorithm in between MLE and policy gradient. Mathematically, RAML shows that MLE and maximum-entropy policy gradient are respectively minimizing KL divergences in opposite directions. Koyamada et al. (2018) thus propose to use the more general $\alpha$-divergence as a combination of the two paradigms. Our framework is developed in a different perspective, reformulates a different and more comprehensive set of algorithms, and leads to new insights in terms of exploration and learning efficiency of the various algorithms. Besides the algorithms discussed in the paper, there are other learning methods for sequence models. For example, Hal Daumé et al. (2009); Leblond et al. (2018); Wiseman & Rush (2016) use a learning-to-search paradigm for sequence generation or structured prediction. Scheduled Sampling (Bengio et al., 2015) and variants (Zhang et al., 2019) adapt MLE by randomly replacing ground-truth tokens with model predictions as the input for decoding the next-step token.

Policy optimization for reinforcement learning is studied extensively in robotic and game environment. For example, Peters et al. (2010) introduce a relative entropy regularization to reduce information loss during learning. Schulman et al. (2015) develop a trust-region approach for monotonic improvement. Dayan & Hinton (1997); Levine (2018); Abdolmaleki et al. (2018) study the policy optimization algorithms in a probabilistic inference perspective. Zhu et al. (2018) combine imitation learning with RL, whose approach is orthogonal to ours and can be plugged into our framework to incorporate imitation reward. The entropy-regularized policy optimization formulation presented here can be seen as a generalization of many of the previous policy optimization methods. Besides, we formulate the framework primarily in the sequence generation context.

## 3  CONNECTING THE DOTS

We first present a generalized formalism of *entropy regularized policy optimization*. The formulation contains a reward function and two weight hyperparameters that define the learning procedure.

Therefore, varying the values of the reward and weights result in a large space of algorithms. We show that several existing popular algorithms, which were originally proposed in distinct perspectives, can all be seen as members in the space. In particular, we reformulate the MLE algorithm in the same policy optimization form, which enables side-by-side comparison between the broad spectrum of algorithms. The resulting unifying view provides new insights into the exploration and computation efficiency, and creates improved learning approaches for sequence prediction.

For clarity, we present the framework in the sequence generation context. The formulations can straightforwardly be extended to other settings such as imitation learning in robotic and game environments, as discussed briefly at the end of this section and also shown in the experiment.

We first establish the basic notations. Let $\boldsymbol{y} = (y_1, \ldots, y_T)$ be the sequence of $T$ tokens. Let $\boldsymbol{y}^*$ be a training example drawn from the empirical data distribution. From the sequence examples, we aim to learn a sequence generation model $p_\theta(\boldsymbol{y}) = \prod_t p_\theta(y_t|\boldsymbol{y}_{1:t-1})$ with parameters $\boldsymbol{\theta}$. Note that generation of $\boldsymbol{y}$ can condition on other factors. For example, in machine translation, $\boldsymbol{y}$ is the sentence in target language and depends on an input sentence in source language. For simplicity of notations, we omit the conditioning factors.

### 3.1 ENTROPY REGULARIZED POLICY OPTIMIZATION (ERPO)

Policy optimization is a family of reinforcement learning (RL) algorithms. Assume a reward function $R(\boldsymbol{y}|\boldsymbol{y}^*) \in \mathbb{R}$ that evaluates the quality of generation $\boldsymbol{y}$ against the true $\boldsymbol{y}^*$. For example, BLEU score (Papineni et al., 2002) can be a reward in machine translation. The general goal of policy optimization is to learn the model $p_\theta(\boldsymbol{y})$ (a.k.a policy)[1] to maximize the expected reward. Previous work develops entropy regularized approaches, which augment the objective with information theoretic regularizers for stabilized training.

We present a generalized variational formulation of ERPO, which, as we show shortly, has the power of subsuming an array of other popular algorithms. Specifically, we introduce a non-parametric variational distribution $q(\boldsymbol{y})$ w.r.t the model $p_\theta(\boldsymbol{y})$. The objective to maximize is as follows:

$$\mathcal{L}(q, \boldsymbol{\theta}) = \mathbb{E}_q\left[R(\boldsymbol{y}|\boldsymbol{y}^*)\right] - \alpha\mathrm{KL}\big(q(\boldsymbol{y})\|p_\theta(\boldsymbol{y})\big) + \beta\mathrm{H}(q), \tag{1}$$

where $\mathrm{KL}(\cdot\|\cdot)$ is the Kullback–Leibler divergence forcing $q$ to stay close to $p_\theta$; $\mathrm{H}(\cdot)$ is the Shannon entropy imposing maximum entropy assumption on $q$; and $\alpha$ and $\beta$ are balancing weights of the respective terms. Intuitively, the objective is to maximize the expected reward under the variational distribution $q$ while minimizing the distance between $q$ and the model $p_\theta$, with maximum entropy regularization on $q$. The above formulation is relevant to and can be seen as a variant of previous policy optimization approaches in RL literature, such as relative entropy policy search (Peters et al., 2010), maximum entropy policy gradient (Ziebart, 2010; Haarnoja et al., 2017), and other work where the variational distribution $q$ is formulated either as a non-parametric distribution as ours (Abdolmaleki et al., 2018; Peters et al., 2010) or parametric one (Schulman et al., 2015; 2017a; Teh et al., 2017).

The objective can be maximized with a standard EM procedure (Neal & Hinton, 1998) that iterates two coordinate ascent steps optimizing $q$ and $\boldsymbol{\theta}$, respectively. At iteration $n$:

$$\begin{aligned} \text{E-step:} \quad & q^{n+1}(\boldsymbol{y}) \propto \exp\left\{\frac{\alpha \log p_{\theta^n}(\boldsymbol{y}) + R(\boldsymbol{y}|\boldsymbol{y}^*)}{\alpha + \beta}\right\}, \\ \text{M-step:} \quad & \boldsymbol{\theta}^{n+1} = \arg\max_\theta \mathbb{E}_{q^{n+1}}\left[\log p_\theta(\boldsymbol{y})\right]. \end{aligned} \tag{2}$$

In the E-step, $q$ has a closed-form solution, which is an energy-based distribution. We can have an intuitive interpretation of its form. First, it is clear to see that if $\alpha \to \infty$, we have $q^{n+1} = p_\theta^n$. This is also reflected in the objective Eq.(1) where a larger weight $\alpha$ encourages $q$ to be close to $p_\theta$. Second, the weight $\beta$ serves as the temperature of the $q$ softmax distribution. In particular, a large temperature $\beta \to \infty$ makes $q$ a uniform distribution, which is consistent with the outcome of an infinitely large maximum entropy regularization in Eq.(1). In the M-step, the update rule can be interpreted as maximizing the log-likelihood of samples from the distribution $q$.

---

[1]In the following, we will use the term "model" and "policy" exchangeably.

**Token-level Formulation**    In the context of sequence generation, it is sometimes more convenient to express the equations at token level (instead of the sequence level), as shown when we devise a new algorithm in the next section. To this end, we decompose $R(\boldsymbol{y}|\boldsymbol{y}^*)$ along the time steps:

$$R(\boldsymbol{y}|\boldsymbol{y}^*) = \sum_t R(\boldsymbol{y}_{1:t}|\boldsymbol{y}^*) - R(\boldsymbol{y}_{1:t-1}|\boldsymbol{y}^*) := \sum_t \Delta R(y_t|\boldsymbol{y}_{1:t-1}, \boldsymbol{y}^*), \tag{3}$$

where $\Delta R(y_t|\boldsymbol{y}^*, \boldsymbol{y}_{1:t-1})$ measures the reward contributed by token $y_t$. The solution of $q$ in Eq.(2) can then be re-written as:

$$q^{n+1}(\boldsymbol{y}) \propto \prod_t \exp\left\{ \frac{\alpha \log p_{\theta^n}(y_t|\boldsymbol{y}_{1:t-1}) + \Delta R(y_t|\boldsymbol{y}_{1:t-1}, \boldsymbol{y}^*)}{\alpha + \beta} \right\}. \tag{4}$$

**The Algorithm Space**    The above ERPO formalism includes three key components, namely, the reward $R$ and the weight hyperparameters $\alpha$ and $\beta > 0$. Variation in these components can result in different procedures of updating the model. In other words, different algorithms in the ERPO family correspond to a point (or a region) in the space spanned by the three components. The following sections visit a set of existing approaches, and connect them to the unifying picture by reformulating their seemingly distinct objectives. Figure 1 illustrates the particular algorithms in the space, clustered by the exploration behavior in learning, of which we will discuss more.

**Softmax Policy Gradient (SPG)**    We first briefly discuss the previous RL algorithms for sequence prediction that fit in the ERPO formalism. SPG (Ding & Soricut, 2017) was originally developed in the perspective of combining the reward $R$ and policy $p_\theta$ to improve sampling quality. The algorithm is equivalent to setting $\beta = 0$ and treating $\alpha > 0$ as the temperature of the energy-based distribution $q(\boldsymbol{y})$. That is, $q(\boldsymbol{y})$ in the E-step of Eq.(2) is now in the form $q(\boldsymbol{y}) \propto p_\theta(\boldsymbol{y}) \exp\{R(\boldsymbol{y}|\boldsymbol{y}^*)/\alpha\}$. The reward $R$ is set to any normal task-specific reward. Note that sampling from $q(\boldsymbol{y})$ (e.g., in the M-step) is typically difficult due to its energy-based form and the fact that the task reward $R$ often does not have particular structures amenable for sampling. We will see in the next section that the MLE algorithm in contrast uses a special reward to avoid the computational difficulty in sampling, at the cost of restricted exploration during training.

We also note the previous work of **Sequence Tutor** (Jaques et al., 2017), which was motivated by the idea of using an MLE-trained policy as a prior to guide the learning of the target policy in an RL framework. The formalism closely resembles SPG, namely $(\alpha > 0, \beta = 0)$, with the exception that the variational distribution $q(\boldsymbol{y})$ in Sequence Tutor is a parameterized model instead of a non-parametric one as in SPG and our more general ERPO formulation.

### 3.2    MLE as a Special Case of ERPO

In this section, we connect the maximum likelihood estimation (MLE) algorithm to the unifying ERPO formalism. Based on the connections, we are able to analyze the learning behavior of MLE from the reinforcement learning perspective in terms of exploration efficiency. We also discuss some well-known variants of the vanilla MLE algorithm, such as RAML and data augmentation.

Due to its simplicity and efficiency, MLE is among the most widely-used approaches in learning sequence generation. It finds the optimal parameter value that maximizes the data log-likelihood:

$$\boldsymbol{\theta}^* = \arg\max_\theta \mathcal{L}_{\text{MLE}}(\boldsymbol{\theta}) = \arg\max_\theta \log p_\theta(\boldsymbol{y}^*). \tag{5}$$

We show that the MLE objective can be recovered from Eq.(2) with a specialized reward and weight values. More concretely, consider a $\delta$-function reward defined as[2]:

$$R_\delta(\boldsymbol{y}|\boldsymbol{y}^*) = \begin{cases} 1 & \text{if } \boldsymbol{y} = \boldsymbol{y}^* \\ -\infty & \text{otherwise.} \end{cases} \tag{6}$$

That is, a sample $\boldsymbol{y}$ receives a valid unit reward only when it matches exactly with the true data, and receives a negative infinite reward in all other cases.

We show that the MLE algorithm is a member of the ERPO family. In particular, the conventional MLE objective is equivalent to setting the ERPO components to $(R = R_\delta, \alpha \to 0, \beta = 1)$. This can

---

[2]For token-level, define $R_\delta(\boldsymbol{y}_{1:t}|\boldsymbol{y}^*) = t/T^*$ if $\boldsymbol{y}_{1:t} = \boldsymbol{y}^*_{1:t}$ and $-\infty$ otherwise, where $T^*$ is the length of $\boldsymbol{y}^*$. Note that the $R_\delta$ value of $\boldsymbol{y} = \boldsymbol{y}^*$ can also be set to any constant larger than $-\infty$.

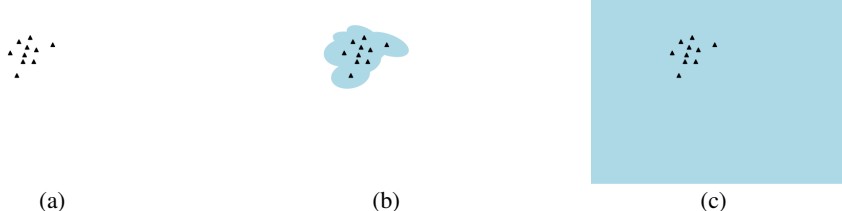

|       |       |       |
|-------|-------|-------|
| (a)   | (b)   | (c)   |

Figure 2: Exploration space exposed to model learning in different algorithms. **(a)**: The effective exploration space of MLE is exactly the set of training examples. **(b)**: Data Noising and RAML use diffused rewards and allow larger exploration space surrounding the training examples. **(c)**: Standard policy optimization such as SPG (section 3.1) basically allows the whole exploration space.

be straightforwardly seen by noting that, with the configuration, the $q(\boldsymbol{y})$ in E-step (Eq.2) reduces to $q(\boldsymbol{y}) = 1$ if $\boldsymbol{y} = \boldsymbol{y}^*$ and 0 otherwise. The M-step is thus in effect maximizing the log-likelihood of the real data examples (Note that the very small $\alpha$ is still $> 0$, making the M-step for maximizing the objective Eq.(1) valid and necessary). With the $\delta$-reward $R_\delta$, any sample $\boldsymbol{y}$ that fails to match the given data $\boldsymbol{y}^*$ exactly will get a negative infinite reward and thus never contribute to model learning.

**Exploration Efficiency**  Reformulating MLE in the unifying ERPO form enables us to directly compare the approach with other RL algorithms. Specifically, the $\delta$-reward has permitted only samples that match training examples, and made invalid any exploration beyond the small set of training data (Figure 2(a)). The extremely restricted exploration at training time results in a brittle model that can easily encounter unseen states and make mistakes in prediction.

On the other hand, however, a major advantage of the $\delta$-reward is that it defines a distribution over the sequence space such that sampling from the distribution is reduced to simply picking an instance from the training set. The resulting samples are ensured to have high quality. This makes the MLE implementation very simple and the computation efficient in practice.

On the contrary, task-specific rewards (such as BLEU) used in standard policy optimization are more diffused than the $\delta$-reward, and thus allow exploration in a broader space with valid reward signals. However, the diffused rewards often do not lead to a distribution that is amenable for sampling as above. The model distribution is thus instead used to propose samples, which in turn can yield low-quality (i.e., low-reward) samples especially due to the huge sequence space. This makes the exploration inefficient or even impractical.

Given the opposite behaviors of the algorithms in terms of exploration and computation efficiency, it is a natural idea to seek a middle ground between the two extremes in order to combine the advantages of both. Previous work has proposed variants of the vanilla MLE from different perspectives. We re-visit some of the popular approaches, and show that they can also be canonicalized in the ERPO framework and enrich our understanding of the learning behaviors.

**Data Noising**  Adding noise to training data is a widely adopted model regularizing technique. Previous work (e.g., Xie et al., 2017) has proposed several data noising strategies in the sequence generation context, such as replacing subsets of tokens with other random words. The resulting noisy data is then used in MLE training. Though previous literature has commonly seen such techniques as a data pre-processing step, we show that the approach can be expressed in the generalized ERPO formulation. Specifically, data noising can be seen as using a *locally relaxed* variant of the $\delta$-reward:

$$R_\delta^{\text{noise}}(\boldsymbol{y}|\boldsymbol{y}^*) = \begin{cases} 1 & \text{if } \boldsymbol{y} = g(\boldsymbol{y}^*), \\ -\infty & \text{otherwise,} \end{cases} \tag{7}$$

where $g$ denotes any transformation operation that returns a new sample as a noisy version of the input raw data $\boldsymbol{y}^*$. With the relaxed reward, data noising locally expands the exploration surrounding the observed training examples (Figure 2(b)). The added exploration at training time can yield a model that is more robust to error at test time.

**Reward-Augmented Maximum Likelihood (RAML)**  RAML (Norouzi et al., 2016) was originally proposed to incorporate task-specific metrics into the MLE training. Formally, it introduces

an exponentiated reward distribution $e(\boldsymbol{y}|\boldsymbol{y}^*) \propto \exp\{R(\boldsymbol{y}|\boldsymbol{y}^*)/\tau\}$, where $R$ is a task reward and $\tau > 0$ is the temperature. The conventional RAML objective is written as:

$$\mathcal{L}_{\text{RAML}}(\boldsymbol{\theta}) = \mathbb{E}_{\boldsymbol{y}\sim e(\boldsymbol{y}|\boldsymbol{y}^*)}\big[\log p_\theta(\boldsymbol{y})\big]. \tag{8}$$

That is, unlike MLE that directly maximizes the data log-likelihood, RAML first perturbs the data proportionally to the reward distribution, and maximizes the log-likelihood of the resulting samples. Similar to how we map MLE to the ERPO formalism, we can align RAML with the unifying form by setting $\alpha \to 0, \beta$ to the temperature $\tau$, and $R$ to the task reward. Compared to the vanilla MLE, the key feature of RAML is the use of task reward instead of the $\delta$-reward, which permits a larger exploration space surrounding the training examples. On the other hand, same as in SPG (section 3.1), sampling from the energy-based distribution with a diffused reward tends to be difficult, and often requires specialized approximations for computational efficiency (e.g., Ma et al., 2017).

**Other Algorithms & Discussions**   The classic policy gradient algorithm (Sutton et al., 2000) has also been used for sequence prediction (e.g., Ranzato et al., 2016). We We show in the appendix that the approach can also be connected to the unifying ERPO with moderate approximations. Ranzato et al. (2016) also proposed a mixing training strategy that anneals from MLE training to policy optimization. We show in the next section that the particular annealing scheme is a special case of the new, more general interpolation algorithm below. We have presented the framework in the context of sequence generation. The formulation can also be extended to other settings. For example, in game environments, $\boldsymbol{y}$ is a sequence of actions and states. The popular imitation learning method GAIL (Ho & Ermon, 2016) uses an adversarially induced reward $R$ from data, and applies standard RL updates to train the policy. The policy update part can be formulated with our framework as standard policy optimization (with $\alpha > 0, \beta = 0$). The new interpolation algorithm described in the next section can also be applied to improve the vanilla GAIL, as shown in the experiments.

Previous work has also studied connections of relevant algorithms. For example, Norouzi et al. (2016); Koyamada et al. (2018) formulate MLE and policy gradient as minimizing the opposite KL divergences between the model and data/reward distributions. Misra et al. (2018) studied an update equation generalizing maximum marginal likelihood and policy gradient. Our framework differs in that we reformulate a different and more comprehensive set of algorithms for sequence prediction, and provide new insights in terms of exploration and its efficiency, which could not be derived from the previous work. Section 2 discusses more related work on sequence prediction learning.

# 4   INTERPOLATION ALGORITHM

The unifying perspective also leads to new algorithms for improved learning. Here, we present an example algorithm that is naturally inspired by the framework.

As in Figure 1, each of the learning algorithms can be seen as a point in the $(R, \alpha, \beta)$ space. Generally, from left to right, the reward gets more diffused and $\alpha$ gets larger, which results in larger sequence space exposed to model training (Figure 2). More exploration in turn also makes the training less efficient due to lower sample quality. We propose an *interpolation* algorithm with the natural idea of starting learning from the most restricted yet efficient algorithm configuration, and gradually expanding the exploration to decrease the training/test discrepancy. The easy-to-hard learning paradigm resembles the curriculum learning (Bengio et al., 2009). As we have mapped the algorithms to the points in the hyperparameter space, the interpolation becomes straightforward, which reduces to simple *annealing* of the hyperparameter values.

Specifically, during training, we would like to anneal from using the restricted $\delta$-reward $R_\delta$ to using task reward, and anneal from sampling (exploring) by only the reward $R$ to sampling by both $R$ and $p_\theta$. Since $R_\delta$ is a $\delta$-function which would make direct function linear combination problematic, we implement the interpolation strategy in the update rule (Eq.2) and use log-sum-exp for mixing. Formally, let $R_{\text{task}}$ denote a task reward. The negative energy of $q(\boldsymbol{y})$ in Eq.(2) (i.e., the exponent inside $\exp\{\cdot\}$) is now replaced with the interpolated term: $\log(\lambda_1 p_\theta + \lambda_2 \exp\{R_{\text{task}}\} + \lambda_3 \exp\{R_\delta\})$. Note that we have re-organized the weight hyperparameters and used the distribution $(\lambda_1, \lambda_2, \lambda_3)$ to carry out the calibration role of $(\alpha, \beta)$. In particular, as training proceeds, we gradually increase $\lambda_1$ and $\lambda_2$ and decrease $\lambda_3$. The formulation of interpolation in effect converts the energy-based model $q(\boldsymbol{y})$ to a mixture of experts, which makes the sampling from $q(\boldsymbol{y})$ easier, and resembles the

| Model | BLEU |
|---|---|
| MLE | $31.99 \pm 0.17$ |
| RAML | $32.51 \pm 0.37$ |
| MIXER | $32.69 \pm 0.09$ |
| MIXER-alike Anneal | $32.65 \pm 0.11$ |
| Self-critic | $32.23 \pm 0.15$ |
| SS | $32.13 \pm 0.14$ |
| **Ours** | $\mathbf{33.35 \pm 0.08}$ |

Table 1: Machine translation results (5-run average $\pm$ std dev). See the text for more details.

| Method | ROUGE-1 | ROUGE-2 | ROUGE-L |
|---|---|---|---|
| MLE | $36.11 \pm 0.21$ | $16.39 \pm 0.16$ | $32.32 \pm 0.19$ |
| RAML | $36.30 \pm 0.04$ | $16.69 \pm 0.20$ | $32.49 \pm 0.17$ |
| Self-critic | $36.48 \pm 0.24$ | $16.84 \pm 0.26$ | $32.79 \pm 0.26$ |
| SS | $36.59 \pm 0.12$ | $16.79 \pm 0.22$ | $32.77 \pm 0.17$ |
| **Ours** | $\mathbf{36.72 \pm 0.29}$ | $\mathbf{16.99 \pm 0.17}$ | $\mathbf{32.95 \pm 0.33}$ |

Table 2: Text summarization results (5-run average $\pm$ std dev).

bang-bang rewarded SPG method as described in (Ding & Soricut, 2017). Besides, similar to (Ding & Soricut, 2017), we adopt the token-level formulation (Eq.4), so that tokens in a sequence can be sampled from different components (i.e., $p_\theta$, $R_{\text{task}}$, and $R_\delta$) in a mixed way.

We provide the pseudo-code of the interpolation algorithm in the appendix. As discussed above, we can also apply the interpolation algorithm in game imitation learning, by plugging it into the GAIL (Ho & Ermon, 2016) framework to replace the standard RL routine for policy update. The annealing schedule in this setting is constrained due to the agent interaction with the environment. Specifically, to generate a trajectory (a sequence of actions and states), we sample the beginning part from data (demonstrations), followed by sampling from either the model or reward. Note that data sampling can happen only before model/reward sampling, because the latter will interact with the environment and result in states that do not necessarily match the data. Similar to sequence generation, we gradually anneal from data sampling to model/reward sampling, and hence increase the exploration until converging to standard RL. Our experiments validate that the easy-to-hard training is superior to the vanilla GAIL which directly applies the hard RL update from the beginning.

It is notable that (Ranzato et al., 2016) also developed an annealing strategy that mixes MLE and policy gradient training. The strategy is essentially the same as the one we apply in the GAIL learning setting. That is, the annealing approach of (Ranzato et al., 2016) is a specialized case of the above more general annealing, using restricted values of $(\lambda_1, \lambda_2, \lambda_3)$ and discrete changes. We provide more discussions in the appendix. The experiment results in section 5 show that our generalized annealing performs better than the restricted approach (Ranzato et al., 2016).

## 5 EXPERIMENTS

We evaluate the interpolation algorithm in the context of both text generation and game imitation learning. Experiments are run with 4 GTX 2080Ti GPUs and 32GB RAM. The link to the code is provided in the submission. We will release the code upon acceptance.

### 5.1 MACHINE TRANSLATION

We use the state-of-the-art neural architecture Transformer (Vaswani et al., 2017) as the base model. The model has 6 blocks, trained with an Adam optimizer with an initial learning rate of 0.001 and the same schedule as in (Vaswani et al., 2017). Batch size is 1,792 tokens. At test time, we use beam search decoding with a beam width of 5 and length penalty 0.6. We use the popular IWSLT2014 (Cettolo et al., 2014) German-English dataset. After proper pre-processing as described in the appendix, we obtain the final dataset with train/dev/test size of around 146K/7K/7K, respectively. The shared de-en vocabulary is of size 73,197 without BPE encoding.

Table 1 shows the test-set BLEU scores of various methods. Besides MLE, RAML, and MIXER (Ranzato et al., 2016) as discussed above, we also compare with other existing approaches such as Scheduled Sampling (SS) (Bengio et al., 2015) and Self-critic (Rennie et al., 2017). (We did not compare with SPG (Ding & Soricut, 2017) as no public code is available.) From the table, we can see the various approaches provide improved performance over the vanilla MLE, as more sufficient exploration is made at training time. Our interpolation algorithm performs best, with significant improvement over the MLE training by 1.36 BLEU points. The results validate our approach that interpolates among the existing algorithms offers beneficial scheduled training. To further study the effect of our generalized annealing versus the MIXER strategy, we compare with "*MIXER-alike*

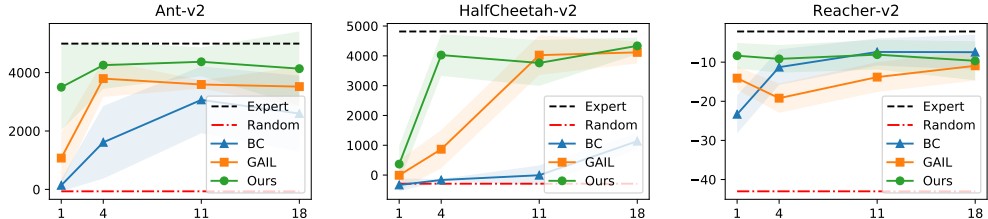

Figure 3: Performance of learned policies. The x-axis is the number of expert demonstrations for training. The y-axis is the average returns. "BC" is Behavior Cloning. "Random" is a baseline taking a random action each time. Results are averaged over 50 runs.

*Aneal*" which uses the same configuration with our interpolation algorithm, except that the annealing is restricted like MIXER. That is, the first portion of tokens in a sequence are all sampled from the data, while the subsequent tokens are sampled from only the model or the task reward. We see that the proposed more generalized annealing is superior to the restricted version. We note that there is other work exploring various network architectures for machine translation (Shankar & Sarawagi, 2019; He et al., 2018), which is orthogonal and complementary to the learning algorithms. It would be interesting to explore the effect of combining the approaches.

## 5.2 TEXT SUMMARIZATION

We use an attentional sequence-to-sequence model (Luong et al., 2015) where both the encoder and decoder are single-layer LSTM RNN. The dimensions of word embedding, RNN hidden state, and attention are all set to 256. We use Adam optimization with an initial learning rate of 0.001 and a batch size of 64. Test time uses beam search decoding with a beam width of 5. Please see the appendix for more configuration details. We use the popular English Gigaword corpus (Graff et al., 2003) for text summarization, and pre-processed the data following (Rush et al., 2015). The resulting dataset consists of 200K/8K/2K source-target pairs in train/dev/test sets, respectively.

Following previous work (Ding & Soricut, 2017), we use the summation of the three ROUGE(-1, -2, -L) metrics as the reward in learning. Table 2 show the results on the test set. The proposed interpolation algorithm achieves the best performance on all three metrics. The RAML algorithm, which performed well in machine translation, falls behind other algorithms in text summarization. In contrast, our method consistently provides the best results.

## 5.3 GAME IMITATION LEARNING

We apply the interpolation algorithm in GAIL (Ho & Ermon, 2016) as described in section 4. Following (Ho & Ermon, 2016), we simulate three environments with MuJoCo (Todorov et al., 2012). Expert demonstrations are generated by running PPO (Schulman et al., 2017b) under the given true reward functions. We then run different imitation learning algorithms with varying numbers of demonstrations. Both the policy and the discriminator are two-layer networks with 128 units each and tanh activations in between.

Figure 3 shows the average returns by the agents. We can see that agents trained with the interpolation algorithm can generally improve over the vanilla GAIL, especially in the presence of small number (e.g., 1 or 4) of demonstrations. This shows that our approach that anneals from the MLE mode to RL mode can make better use of data examples, and steadily achieve better performance in the end. We present the learning curves of the algorithms in the appendix.

## 6 CONCLUSIONS

We have presented a unifying perspective of a variety of learning algorithms for sequence prediction problems. The framework is based on a generalized entropy regularized policy optimization formulation, and we show the distinct algorithms are equivalent to specifying the reward and weight hyperparameters. The new consistent treatment provides systematic understanding and comparison across the algorithms, and inspires further improved learning. The proposed interpolation algorithm shows consistent improvement in machine translation, text summarization, and game imitation learning.

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

# A  APPENDIX

## A.1  POLICY GRADIENT & MIXER

Ranzato et al. (2016) made an early attempt to address the exposure bias problem by exploiting the policy gradient algorithm (Sutton et al., 2000). Policy gradient aims to maximizes the expected reward:

$$\mathcal{L}_{PG}(\boldsymbol{\theta}) = \mathbb{E}_{p_\theta}\left[R_{PG}(\boldsymbol{y}|\boldsymbol{y}^*)\right], \tag{9}$$

where $R_{PG}$ is usually a common reward function (e.g., BLEU). Taking gradient w.r.t $\boldsymbol{\theta}$ gives:

$$\nabla_\theta \mathcal{L}_{PG}(\boldsymbol{\theta}) = \mathbb{E}_{p_\theta}\left[R_{PG}(\boldsymbol{y}|\boldsymbol{y}^*)\nabla_\theta \log p_\theta(\boldsymbol{y})\right]. \tag{10}$$

We now reveal the relation between the ERPO framework we present and the policy gradient algorithm. Starting from the M-step of Eq.(2) and setting ($\alpha = 1, \beta = 0$) as in SPG (section **??**), we use $p_{\theta^n}$ as the proposal distribution and obtain the importance sampling estimate of the gradient (we omit the superscript $n$ for notation simplicity):

$$\begin{aligned}
\mathbb{E}_q\left[\nabla_\theta \log p_\theta(\boldsymbol{y})\right] &= \mathbb{E}_{p_\theta}\left[\frac{q(\boldsymbol{y})}{p_\theta(\boldsymbol{y})}\nabla_\theta \log p_\theta(\boldsymbol{y})\right] \\
&= 1/Z_\theta \cdot \mathbb{E}_{p_\theta}\left[\exp\{R(\boldsymbol{y}|\boldsymbol{y}^*)\} \cdot \nabla_\theta \log p_\theta(\boldsymbol{y})\right],
\end{aligned} \tag{11}$$

where $Z_\theta = \int_y \exp\{\log p_\theta + R\}$ is the normalization constant of $q$, which can be considered as adjusting the step size of gradient descent.

We can see that Eq.(11) recovers Eq.(10) if we further set $R = \log R_{PG}$, and omit the scaling factor $Z_\theta$. In other words, policy gradient can be seen as a special instance of the general ERPO framework with ($R = \log R_{PG}, \alpha = 1, \beta = 0$) and with $Z_\theta$ omitted.

The **MIXER** algorithm (Ranzato et al., 2016) incorporates an annealing strategy that mixes between MLE and policy gradient training. Specifically, given a ground-truth example $\boldsymbol{y}^*$, the first $m$ tokens $\boldsymbol{y}_{1:m}^*$ are used for evaluating MLE loss, and starting from step $m+1$, policy gradient objective is used. The $m$ value decreases as training proceeds. With the relation between policy gradient and ERPO as established above, MIXER can be seen as a specific instance of the proposed interpolation algorithm (section 4) that follows a restricted annealing strategy for token-level hyperparameters ($\lambda_1, \lambda_2, \lambda_3$). That is, for $t < m$ in Eq.4 (i.e.,the first $m$ steps), ($\lambda_1, \lambda_2, \lambda_3$) is set to $(0, 0, 1)$ and $c = 1$, namely the MLE training; while for $t > m$, ($\lambda_1, \lambda_2, \lambda_3$) is set to $(0.5, 0.5, 0)$ and $c = 2$.

## A.2  INTERPOLATION ALGORITHM

Algorithm 1 summarizes the interpolation algorithm described in section 4.

## A.3  EXPERIMENTAL SETTINGS

### A.3.1  DATA PRE-PROCESSING

For the machine translation dataset, we follow (Ma et al., 2017) for data pre-processing.

In text summarization, we sampled 200K out of the 3.8M pre-processed training examples provided by (Rush et al., 2015) for the sake of training efficiency. We used the refined validation and test sets provided by (Zhou et al., 2017).

In the game imitation learning task, we randomly sample 50 state-action pairs in each trajectory as demonstrations. Every training iteration, we collect at least 2,048 state-action pairs, and we train 1,000 iterations for every model in every environment.

---

**Algorithm 1** Interpolation Algorithm

---

1: Initialize model parameter $\boldsymbol{\theta}$ and weights $\boldsymbol{\lambda} = (\lambda_1, \lambda_2, \lambda_3)$
2: **repeat**
3:     Get training example $\boldsymbol{y}^*$
4:     **for** $t = 0, 1, \ldots, T$ **do**
5:         Sample $z \in \{1, 2, 3\} \sim (\lambda_1, \lambda_2, \lambda_3)$
6:         **if** $z = 1$ **then**
7:             Sample token $y_t \sim \exp\{c \cdot \log p_\theta(y_t | \boldsymbol{y}_{1:t-1})\}$
8:         **else if** $z = 2$ **then**
9:             Sample token $y_t \sim \exp\{c \cdot \Delta R(y_t | \boldsymbol{y}_{1:t-1}, \boldsymbol{y}^*)\}$
10:         **else**
11:             Sample token $y_t \sim \exp\{c \cdot \Delta R_\delta\}$, i.e., set $y_t = y_t^*$
12:         **end if**
13:     **end for**
14:     Update $\boldsymbol{\theta}$ by maximizing the log-likelihood $\log p_\theta(\boldsymbol{y})$
15:     Anneal $\boldsymbol{\lambda}$ by increasing $\lambda_1$ and $\lambda_2$ and decreasing $\lambda_3$
16: **until** convergence

---

### A.3.2   ALGORITHM SETUP

For RAML (Norouzi et al., 2016), we use the sampling approach (n-gram replacement) by (Ma et al., 2017) to sample from the exponentiated reward distribution. For each training example we draw 6 and 10 samples in machine translation and text summarization tasks, respectively.

For Scheduled Sampling (SS) (Bengio et al., 2015), we tested various annealing schedules and report the best-performing one, namely inverse-sigmoid decay. The probability of sampling from model $\epsilon_i = k/(k + \exp(i/k))$, where $k$ is a hyperparameter controlling the speed of convergence, which is set to 4000 and 600 in the machine translation and text summarization tasks, respectively. We would like to note that SS does not fit into our formulation, because, in SS, model-generated tokens are only used as model inputs instead of the targets of which the likelihood is maximized. For example, at time step $t$, even though token $\hat{y}_t$ generated by the model is used as an input to the next step, the loss associated with step $t$ is still $\log p_\theta(y_t^* | \text{prev tokens})$ where $y_t^*$ is the true token. This differs from our formulation which maximizes the likelihood of $\hat{y}_t$.

For the proposed interpolation algorithm, after MLE pre-training, we initialize the weights as $(\lambda_1, \lambda_2, \lambda_3) = (0.12, 0.16, 0.72)$. Every 4 epochs, we increase $\lambda_1$ by 0.12 and $\lambda_2$ by 0.16 while decreasing $\lambda_3$ by 0.28.

We did MLE pretraining for all comparison methods for the same number of steps. We found pretraining is necessary for Self-critic, and is helpful for RAML and SS.

### A.3.3   LEARNING CURVES OF GAIL EXPERIMENTS

Figure 4 presents the learning curves of different algorithms in the GAIL experiments.

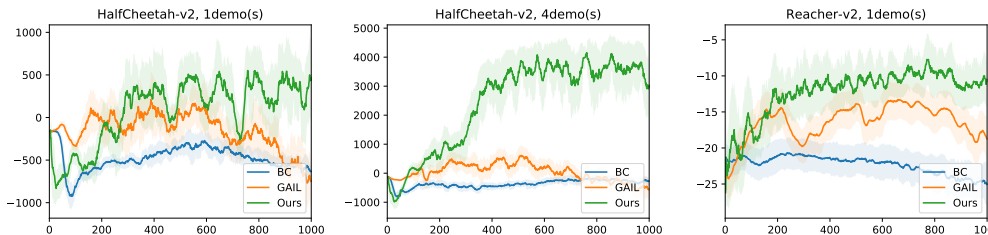

Figure 4: Learning curves of GAIL experiments. The $x$-axis is the number of training iterations. The $y$-axis is the returns. GAIL performance drops in the late stage, as also observed in previous work (e.g., Figure 1 of (Wu et al., 2019))

