# OpenReview forum: "Connecting the Dots Between MLE and RL for Sequence Prediction"
_ICLR.cc/2020/Conference — Reject_

### Official Review · AnonReviewer3 · 2019-10-18
**Official Blind Review #1723**

**Rating:** 3

**Review:**

This paper claims to propose a general entropy regularized policy optimization paradigm. MLE and RL are special cases of this training paradigm. Paper is well written, and the experimental results are convincing enough.
However, there are still some minor problems in the paper. For the optimization framework ERPO (shown in Equation 1), it consists of three parts, a cross-entropy term (Shannon entropy), a $p,q$ KL divergence term, and a reinforcement learning reward loss item. From the framework point of view, it is not like the author claim that is supposed to present a general optimization framework, including various optimization algorithms. Instead, it is just a combined loss through weight control and the selection of corresponding functions. It may not really theoretically work to unify various types of optimization algorithms for general cases, let alone claiming that this is a general optimization algorithm framework.

For the interpolation algorithm (I regard this is the true technical contribution of this paper), the authors used an annealing mechanism to use different weights and functions at different stages of training. The essence is that after MLE pre-training, different optimization algorithms are used in different stages, and this should be the focus of the article. The annealing settings used is only introduced in the appendix simply. Without more comparison experiments, we cannot clearly get the conditions for the annealing algorithm to be effective and ineffective.

For the title of connecting the dots between MLE and RL, this paper did not do so, MLE and RL are only used collaboratively, and this has also been mentioned in previous work.

typo
Page 6 Paragraph “Other Algorithms & Discussions”: We We show in the appendix… -> We show in the appendix…


**Experience Assessment:**

I have published one or two papers in this area.

**Review Assessment: Checking Correctness Of Derivations And Theory:**

I assessed the sensibility of the derivations and theory.

**Review Assessment: Checking Correctness Of Experiments:**

I carefully checked the experiments.

**Review Assessment: Thoroughness In Paper Reading:**

I read the paper thoroughly.

---

> ### Author Response · Authors · 2019-11-15
> **Response to Official Blind Review #1723**
>
> Thanks for the comments! We’d like to clarify that this paper aims to reformulate the various algorithms and distill them into a single common formulation. The common formulation is governed by the reward function and two weight hyperparameters, and thus defines a *family* of sequence prediction algorithms. Changing the specifications of the three factors (i.e., reward and weights) leads to different specific algorithms. It’s indeed novel and non-trivial to reformulate these apparently distinct algorithms and discover the common underlying formulation. We will update the presentation (as also suggested by R#1) accordingly to make this contribution clearer.
>
> The interpolation algorithm is an immediate production of the discovered common formulation, as it’s a natural idea, once we see the common formulation, to anneal the three governing factors to “interpolate” between the specific algorithms in the family. We add the discussion of the annealing settings in the main paper. It’s also worth noting that the interpolation algorithm is just one of the many possible ways of taking advantage of the common formulation. For example, another natural idea would be to find (e.g., through hyperparameter optimization) the best configuration of the three governing factors in the common formulation, which is equivalent to finding the best optimization algorithm in the whole family and use it to learn sequence prediction. This shows the advantages of having the common formulation. We will make this clearer in the revised version.
>
> This paper is the first to discuss the extensive set of algorithms (MLE, RAML, Data Noising, policy gradient, etc) jointly and find their common denominator. The resulting interpolation algorithm can also be seen as a generalization of previous approaches, such as MIXER, that “uses MLE and RL collaboratively”. (As discussed in the paper, MIXER can be seen as using a restricted annealing strategy in the proposed interpolation algorithm). The generalized interpolation algorithm also outperforms MIXER.
>
> We will fix all typos and do proofread. Thanks for pointing this out.

---

### Official Review · AnonReviewer1 · 2019-10-23
**Official Blind Review #1**

**Rating:** 3

**Review:**

This submission belongs to the field of sequence modelling. In particular, this submission presents a unified view on a range of training algorithms including maximum likelihood (ML) and reinforcement learning (RL). The unified view presented I believe is interesting and could be of interest to a large community. Unfortunately this submission has two issues 1) presentation and 2) experimental validation.

I find it peculiar that an objective function that features ML and variants of RL as special cases called ERPO is proposed by statement. I find it more likely that it came out by analysing ML, the variants of RL and other commonly used objective functions, noticing similarities between them and then formulating a function that would render all above as special cases. Had the order been different this submission would have been much more analytical and interesting to read.

I find experimental results a bit limited and not entirely conclusive as it seem that MT provides the only strong experimental evidence. I find quite hard to interpret the significance of difference, for instance, between 36.72 and 36.59 in ROUGE-1.

**Experience Assessment:**

I have published in this field for several years.

**Review Assessment: Checking Correctness Of Derivations And Theory:**

I carefully checked the derivations and theory.

**Review Assessment: Checking Correctness Of Experiments:**

I carefully checked the experiments.

**Review Assessment: Thoroughness In Paper Reading:**

I read the paper thoroughly.

---

> ### Author Response · Authors · 2019-11-15
> **Response to Official Blind Review #1**
>
> Thanks for the great suggestion of adjusting the order of presentation! We agree that it can be clearer and more analytical by reaching the final formulation after visiting each individual algorithms. We will update the presentation in the revised version accordingly.
>
> Besides MT, the improvement in game imitation learning is indeed reasonably significant, especially when the number of expert demonstrations is small:
>   ** On HalfCheetah-v2 with 4 demonstrations, our approach achieves *3000 higher* reward than GAIL; On Ant-v2, our approach achieves *800 higher* reward on average.
>   ** The improvement level over GAIL is comparable or stronger than that in other papers (e.g., Fig.2 of NeurIPS-2018 https://arxiv.org/pdf/1805.08336.pdf; Fig.1 of ICML-2019 https://arxiv.org/pdf/1901.09387.pdf) which proposed to improve GAIL in different aspects orthogonal to ours.
>
> The improvement on summarization is relatively moderate, partially because the output sequences of the task are short (8.2 tokens on average), while the proposed approach is designed for generating longer sequences (e.g., in the MT dataset, the output sequences contain 18.5 tokens on average).

---

### Official Review · AnonReviewer2 · 2019-10-24
**Official Blind Review #2**

**Rating:** 6

**Review:**

This paper presents a formalism of entropy regularized policy optimization. They also show that various policy gradients algorithms can be reformulated as special instances of the presented novel formalism. The only difference between them being the reward function and two weight hyperparameters. Further, the paper proposes an interpolation algorithm, which, as training proceeds, gradually expands the exploration of space by annealing the reward function and the weight hyperparameters. Experiments on text generation tasks and game imitation learning show superior performance over previous methods.

Overall, the paper is well written and the derivations and intuitions sound good. I appreciate the overall effort of the paper and the thorough experiments to validate the proposed interpolation algorithm, results seem not significant for text summarization. Hence, I suggest a week accept for this paper.

Arguments:
1) From Table 1 and Table 2, the proposed approach has the lowest variance on machine translation and the quiet opposite on the text summarization (i.e., it has high variance). Any thoughts on this? This also suggests to conduct experiments on ablating the variance in the training for various policy gradient approaches include the proposed one.

2) Results seem not significant on the summarization tasks. Any thoughts on choosing this particular task? Why not try image captioning where most of these policy gradient approaches have been applied.


**Experience Assessment:**

I have published one or two papers in this area.

**Review Assessment: Checking Correctness Of Derivations And Theory:**

I assessed the sensibility of the derivations and theory.

**Review Assessment: Checking Correctness Of Experiments:**

I carefully checked the experiments.

**Review Assessment: Thoroughness In Paper Reading:**

I read the paper at least twice and used my best judgement in assessing the paper.

---

> ### Author Response · Authors · 2019-11-15
> **Response to Official Blind Review #2**
>
> Thanks for the valuable and encouraging comments.
>
> (1) Thanks for the great question and suggestion! The average output length in the machine translation dataset is 18.5. That is, each target sequence contains 18.5 tokens on average. In contrast, the length is 8.2 in the text summarization dataset. We speculate these different output lengths lead to the different variances and improvement --- the proposed approach by design brings more benefits when the output sequences are longer (e.g., MT) by combating the compounding error. The approach thus gets relatively significant improvement and lower variance compared to the other methods on MT; while on summarization, the approach achieves moderate improvement and does not reduce the variance. We will add more analysis and discussions in the revised version.
>
> (2) Thanks for the suggestion! As above, the improvement on the summarization task is moderate due to the short output sequences in the dataset. We will conduct experiments on image captioning and add the results and analysis.

---

### Decision · Program_Chairs · 2019-12-19

**Decision:**

Reject

**Comment:**

The authors construct a weighted objective that subsumes many of the existing approaches for sequence prediction, such as MLE, RAML, and entropy regularized policy optimization. By dynamically tuning the weights in the objective, they show improved performance across several tasks.

Although there were no major issues with the paper, reviewers generally felt that the technical contribution is fairly incremental and the empirical improvements are limited. Given the large number of high-quality submissions this year, I am recommending rejection for this submission.